# Decouple Searching from Training: Scaling Data Mixing via Model Merging for Large Language Model Pre-training

Shengrui Li [1]   Fei Zhao [1]   Kaiyan Zhao [1 2]   Jieying Ye [1]   Haifeng Liu [1]   Fangcheng Shi [1]   Zheyong Xie [1]   Yao Hu [1]
Shaosheng Cao [1 3]

## Abstract

Determining an effective data mixture is a key factor in Large Language Model (LLM) pre-training, where models must balance general competence with proficiency on hard tasks such as math and code. However, identifying an optimal mixture remains an open challenge, as existing approaches either rely on unreliable tiny-scale proxy experiments or require prohibitively expensive large-scale exploration. To address this, we propose Decouple Searching from Training Mix (DeMix), a novel framework that leverages model merging to predict optimal data ratios. Instead of training proxy models for every sampled mixture, DeMix trains component models on candidate datasets at scale and derives data mixture proxies via weighted model merging. This paradigm decouples search from training costs, enabling evaluation of unlimited sampled mixtures without extra training burden and thus facilitating better mixture discovery through more search trials. Extensive experiments demonstrate that DeMix breaks the trade-off between sufficiency, accuracy and efficiency, obtaining the optimal mixture with higher benchmark performance at lower search cost. Additionally, we release the DeMix Corpora, a comprehensive 22T-token dataset comprising high-quality pre-training data with validated mixtures to facilitate open research. Our code and DeMix Corpora is available at https://github.com/Lucius-lsr/DeMix.

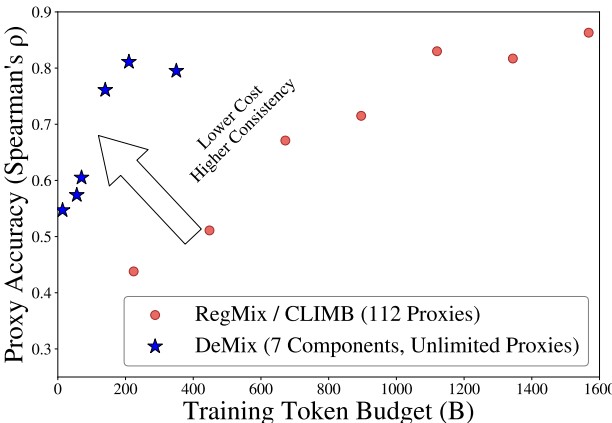

*Figure 1.* Methods such as RegMix and CLIMB require extensive proxies. Scaling each proxy leads to unaffordable overall budget. While our DeMix only require a few component models to merge unlimited training-free proxies.

## 1. Introduction

Large Language Models (LLMs) have achieved remarkable success across a wide range of domains (Shao et al., 2024; Guo et al., 2025; Kimi Team et al., 2025; Bai et al., 2025), largely driven by the massive pre-training (Gemini Team et al., 2023; Achiam et al., 2023). Beyond scale alone, the composition of the pre-training corpus plays a critical role in shaping model capabilities (Feng et al., 2024; Basant et al., 2025; Blakeman et al., 2025).

However, optimizing the data mixture for pre-training remains challenging, as it requires balancing general-purpose language abilities with strong performance on complex tasks such as mathematical reasoning and code generation (Cobbe et al., 2021a; Chen et al., 2021; Wei et al., 2022).

A commonly adopted strategy for data mixture selection is to conduct limited large-scale proxy experiments, where mid-sized models (e.g., 8B) are trained on sampled data mixtures with substantial token budgets (e.g., 100B) (Li et al., 2025; Nie et al., 2025; Blakeman et al., 2025). While such proxies can provide relatively accurate signals, they remain

[1]NLP Team, Xiaohongshu Inc., Shanghai, China [2]The University of Tokyo, Tokyo, Japan [3]Tsinghua University, Beijing, China. Correspondence to: Shaosheng Cao <caoshaosheng@xiaohongshu.com>.

*Proceedings of the 43rd International Conference on Machine Learning*, Seoul, South Korea. PMLR 306, 2026. Copyright 2026 by the author(s).

computationally expensive and are insufficient for systematically identifying optimal data mixtures. In contrast, recent lines of work like RegMix (Liu et al., 2024) and CLIMB (Diao et al., 2025) aim to fully automate the process of searching the optimal data mixture ratios. These approaches rely on extensive tiny-scale proxy experiments with smaller models and reduced data budgets to train regression-based predictors that map data mixture to loss or downstream performance. However, the validity of these lightweight proxies has been increasingly questioned (Allal et al., 2025b). Due to the substantial capability gap between proxy and target models, these automated strategies often fail to generalize to complex tasks such as math and code.

Moreover, despite the abundance of domain-specific pre-training data, there is a notable lack of benchmarked corpora with validated data mixture ratios that can be directly reused for large-scale pre-training.

To address these challenges, we propose **De**couple Search-ing from Training **Mix** (**DeMix**), together with the pre-training dataset **DeMix Corpora**. Instead of training a large number of proxy models under different data mixture ratios, DeMix decouples mixture search from proxy training by leveraging model merging: *component models* are merged to synthesize an effectively unlimited number of proxy models at virtually no additional cost. Specifically, we first train a set of component models at scale, each corresponding to a candidate dataset. We then construct proxy models via weighted model merging over these trained component models, where the merging weights represent target data mixture ratios. To evaluate the fidelity of model-merged proxy models, we compare them with reference models that are trained directly on sampled data mixtures with massive tokens. We find that these model-merged proxies exhibit substantially higher ranking consistency with these reference models than proxies obtained through traditional small-scale training.

As shown in Figure 1, DeMix achieves substantially higher proxy accuracy than training-based proxy models under the same limited budget, and reach comparable accuracy with approximately $6\times$ less computation budget (200 v.s. 1200). Thus, DeMix simultaneously achieves sufficiency (unlim-ited proxy models), accuracy (faithful proxy performances) and efficiency (fixed token budgets).

Based on these merged proxy models, we predict the op-timal data mixture via regression-based methods and ap-ply it to sample 50B tokens for pre-training a 1.7B model. Through extensive experiments across multiple benchmarks covering general language understanding, mathematical rea-soning, and code generation, we observe that DeMix sig-nificantly outperforms other state-of-the-art data mixture methods such as Regmix and CLIMB while requiring less computational budget. To summarize, our contributions are as follows:

- We propose DeMix, a framework that efficiently de-couples data mixture search from model training by constructing proxy models through weighted merging of component models.

- We demonstrate that model-merging proxies faithfully preserve the performance ordering of reference mod-els trained on real data mixtures, providing a reliable signal for mixture selection.

- We release **DeMix Corpora**, a 22T high-quality, large-scale dataset with validated mixtures that can be di-rectly used for LLM pre-training.

**Conflict of Interest Disclosure.** The authors declare no financial conflicts of interest.

## 2. Method

The overall pipeline of DeMix is depicted in Figure 2 and is structured into four sequential phases. In this section, we first briefly describe how candidate datasets are obtained through rigorous filtering. Second, we detail the preparation of component models. Third, we then present our approach for constructing proxy models and discuss the feasibility of using merged models as proxies in our setting. Finally, we introduce an iterative mixture strategy designed to further enhance performance.

### 2.1. Dataset Preprocessing

We first collect large-scale data from a variety of sources, including general-domain corpora, mathematical datasets, and code collections. We then apply rigorous data clean-ing, which consists of deduplication, perplexity filtering, FastText filtering and so on. Finally, we perform data-level evaluation and categorize the cleaned corpus into multiple candidate datasets, each representing a distinct data source or domain for subsequent mixture optimization. More de-tails on data preprocessing can be found in Appendix A.

### 2.2. Component Model Preparation

Given the $N$ candidate datasets, we can construct $N$ com-ponent models, each trained individually on one of the $N$ candidate datasets. To ensure the robust performance of these component models, we implement a two-step training protocol. First, all component models are initialized from a shared base model trained from scratch on a general-purpose dataset $D_{\text{base}}$, which provides foundational language capa-bilities (denoted as the Base Model in Figure 2). Second, each component model is further trained on its correspond-ing domain-specific candidate dataset (e.g., mathematics or code), mixed with general data at a fixed ratio $\beta$. This proce-dure encourages each component model to specialize in its

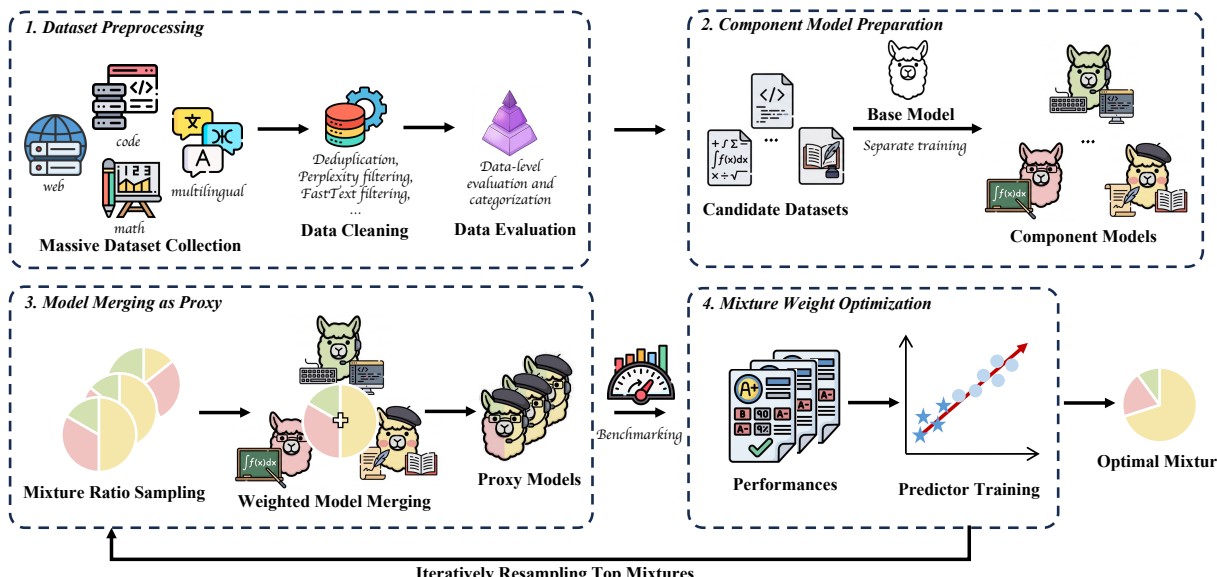

*Figure 2.* Pipeline for DeMix. After (1) cleaning and categorizing massive data, (2) component models are trained on individual candidate datasets. Instead of large-scale training for every ratio, (3) weighted model merging serves as a computationally efficient proxy to estimate performance for various mixture ratios. Finally, (4) a predictor is trained on the benchmarked proxy models to regress the relationship between mixing ratios and performance, utilizing iterative resampling to converge on the optimal mixture.

target domain while retaining general language competence, enabling them to effectively serve as building blocks for subsequent model merging and data mixture optimization.

### 2.3. Model Merging as Proxy

We first briefly explain why model merging can be used to construct proxy models instead of training new proxy models for each sampled data mixture.

Building upon the former sections, let $\Theta_{\text{base}} \in \mathbb{R}^d$ denote the parameter vector of a pre-trained base model. Consider a collection of $N$ candidate datasets $\{D_1, D_2, \ldots, D_N\}$, we define a training operator $\mathcal{T} : \mathcal{D} \times \mathbb{R}^d \to \mathbb{R}^d$, where $\mathcal{T}(D, \Theta_{\text{base}})$ results in the model parameters after training on dataset distribution $D$ initialized from $\Theta_{\text{base}}$. For any dataset $D_i$, the *parameter update vector* (or weight delta) can be defined as (Yang et al., 2024):

$$\Delta(D_i) \triangleq \mathcal{T}(D_i, \Theta_{\text{base}}) - \Theta_{\text{base}}. \quad (1)$$

Consequently, the individually trained component model corresponds to parameters $\Theta_i = \Theta_{\text{base}} + \Delta(D_i)$.

The objective of data mixing is to identify an optimal mixture distribution (Xie et al., 2023):

$$\mathcal{D}_{\text{mix}} = \sum_{i=1}^{N} \alpha_i D_i, \quad \text{with } \alpha_i \geq 0 \text{ and } \sum_{i=1}^{N} \alpha_i = 1, \quad (2)$$

so that the resulting model parameters:

$$\Theta_{\text{mix}} = \mathcal{T}(\mathcal{D}_{\text{mix}}, \Theta_{\text{base}}). \quad (3)$$

achieve the best performance across the target benchmarks.

To formalize the connection between data mixing and model merging, we first state the relevant constraints.

In practice, the magnitude of parameter updates $\Delta(D)$ remains relatively small compared to the initialization scale (Gueta et al., 2023; Wu et al., 2025). Formally, for any dataset $D$ in our context, we define (Wu et al., 2025):

$$\delta = \frac{\sum |\mathcal{T}(D, \Theta_{\text{base}}) - \Theta_{\text{base}}|}{\sum |\mathcal{T}(D, \Theta_{\text{base}})| + \sum |\Theta_{\text{base}}|} \ll 1. \quad (4)$$

In our experiments, $\delta$ is approximately 10%, which satisfies this small-update assumption. Empirical studies have shown that as long as $\delta \ll 1$, the arithmetic sum of weight deltas from models trained on separate datasets closely approximates the weight delta obtained by training on their union (Qin et al., 2022; Wu et al., 2025; Lin et al., 2025b):

$$\Delta(D_i \cup D_j) \approx \Delta(D_i) + \Delta(D_j), \quad (5)$$

which indicates that the model parameters trained on a weighted mixture of datasets $\mathcal{D}_{\text{mix}} = \sum \alpha_i D_i$ can be approximated by the weighted average of parameters $\sum_{i=1}^{N} \alpha_i \Theta_i$ trained on separate datasets. We further validate the effectiveness of this approximation in our setting through an additional experiment, reported in Appendix D.

Based on this proposition, merging the component models $\{\Theta_i\}$ at a specific ratio $\{\alpha_i\}$ can obtain a proxy model for any real $\Theta_{\text{mix}}$ trained on $\{\alpha_i D_i\}$. The merged model $M_{\text{mix}}^j$

is calculated as:

$$M_{\text{mix}}^j = \sum_{i=1}^{N} \alpha_i^j \Theta_i, \qquad (6)$$

where $\Theta_i$ represents the component model of the corresponding candidate dataset $D_i$.

## 2.4. Mixture Weight Optimization

With an accurate proxy computed as $M_{mix}^j = \sum_{i=1}^{N} \alpha_i^j \Theta_i$, we can search for the optimal mixture through iterative optimization of the mixture weights. Note that inference on any merged proxy model $M_{mix}^j$ can be performed directly, incurring no additional training cost.

We then define the average ranking of the merged model across a suite of benchmarks covering general language understanding, mathematical reasoning, and code generation as the gold-standard evaluation signal. Ranking-based evaluation is chosen for its robustness to scale mismatches and its direct relevance to mixture selection.

Next, we perform the following steps for iterative prediction of the optimal mixture ratio:

- Step 1: Randomly sampling a large set of mixture weights ratios $\{\alpha_i^j\}$ uniformly from the simplex.

- Step 2: For each sampled weight ratio $\{\alpha_i^j\}$, we construct a proxy model by weighted merging of the component models using Equation 6 and evaluate it on the benchmark suite to obtain its average ranking score $r^j$.

- Step 3: Using the collected pairs $(\alpha_i^j, r^j)$, we train a predictor $f$ that maps mixture weights to ranking scores. In practice, we follow Liu et al. (2024) and adopt LightGBM (Ke et al., 2017) as the regression model.

- Step 4: The trained predictor $f$ is then used to score a large number of newly sampled mixture ratios. We select the top ratios according to the predicted $r^j$ and iteratively execute Step 2-4 three times, thereby refining the predictions for the high-ranking ratios.

After the final iteration, we sample a large number of mixture ratios using the trained predictor and select the top-ranked candidates. The final optimal mixture ratio is computed as the average of these candidates, which defines our final mixed dataset for pre-training.

## 3. Experimental Settings

**Models**   In our experiments, we use Qwen3-1.7B (Yang et al., 2025) as the backbone to determine the optimal mixture, and further validate it on both Qwen3-1.7B and Qwen3-8B models. To maintain fundamental capabilities, before

*Table 1.* GPU (H800) hour cost for training and benchmarking. The cost of one benchmarking run is equivalent to training 0.013B tokens.

| Budget / Proxy | Training | Benchmarking |
|:---:|:---:|:---:|
| 2B | 46 GPUh | 0.3 GPUh |
| 0.013B | 0.3 GPUh | 0.3 GPUh |

training on the candidate datasets, each component model is trained from scratch on 50 billion tokens of general data.

**Implementation Details**   For the candidate dataset, we consistently set the data mixing ratio $\beta$ to 0.5 throughout our experiments. In training the component models, we employ a global batch size of 512 and a sequence length of 8192, with an initial learning rate of 3e-4 optimized via a cosine learning rate schedule that retains a minimum of 20% of the initial learning rate. For the model merging procedure, we adopt a straightforward yet effective weighted linear merging strategy. In the iterative prediction process, given a proxy budget of 112, we sample 64, 32, and 16 mixtures in each respective iteration, and subsequently average the top 128 mixtures to derive the final optimal mixture configuration. For the LightGBM model, we set the learning rate to 0.02 and the number of iterations to 300. For model evaluation, we utilize the OpenCompass benchmark (OpenCompass Contributors, 2023).

### 3.1. Benchmark

For evaluation, we adopt ARC-E (Clark et al., 2018), HellaSwag (Zellers et al., 2019), WinoGrande (Sakaguchi et al., 2021), PIQA (Bisk et al., 2020), and SIQA (Sap et al., 2019) as general benchmarks; HumanEval (Chen et al., 2021) and MBPP (Austin et al., 2021) as code benchmarks; and GSM8K (Cobbe et al., 2021b) and MATH (Hendrycks et al., 2021) as math benchmarks. As shown in Table 1, the cost of benchmarking is negligible compared to the training cost. To facilitate comparison, we convert the GPU hour consumption and find that the cost of one benchmarking run is equivalent to training 0.013B tokens.

### 3.2. Baselines

We adopt the state-of-the-art methods including RegMix (Liu et al., 2024) and CLIMB (Diao et al., 2025) as our baselines. We modify most configurations of RegMix and CLIMB—including the model, data, and training budget—to align with our setup. We use 112 proxies by default following the original setting in CLIMB (64+32+16), and we also experiment with other proxy counts. The detailed settings are available in Appendix B. We exclude earlier inferior methods, including DoReMi (Xie et al., 2023) and Rho Loss (Mindermann et al., 2022), as they depend on

*Table 2.* Comparison of training costs and proxy accuracy between DeMix and training-based proxies. † is the benchmarking cost derived from equal GPU hour. The best scores for DeMix and trained proxy are bolded.

| Method | Train Cost (B) ↓ | | | | Spearman's $\rho$ ↑ | | | | Top 25% Spearman's $\rho$ ↑ | | | | Capability Recovery ↑ | | | |
| | Total Budget | Pre-Cost | No. Proxies | Budget / Proxy | General | Code | Math | Macro Avg. | General | Code | Math | Macro Avg. | General | Code | Math | Macro Avg. |
|---|---|---|---|---|---|---|---|---|---|---|---|---|---|---|---|---|
| Trained Proxy (RegMix/CLIMB) | 224 | - | 112 | 2 | 0.12 | 0.60 | 0.85 | 0.53 | 0.17 | 0.63 | -0.20 | 0.20 | 0.98 | 0.76 | 0.57 | 0.77 |
| | 448 | - | 112 | 4 | -0.24 | 0.81 | 0.96 | 0.51 | -0.27 | 0.67 | 0.73 | 0.38 | 0.98 | 0.80 | 0.63 | 0.80 |
| | 896 | - | 112 | 8 | 0.41 | 0.77 | 0.97 | 0.71 | 0.40 | 0.43 | 0.57 | 0.47 | 0.99 | 0.83 | 0.69 | 0.83 |
| | 1344 | - | 112 | 12 | 0.54 | 0.94 | 0.97 | **0.82** | 0.03 | 0.70 | 0.97 | **0.57** | 0.99 | 0.89 | 0.74 | **0.87** |
| DeMix (Ours) | 15 | 2×7 | 112 | 0.01† | 0.25 | 0.51 | 0.88 | 0.55 | -0.17 | 0.01 | 0.97 | 0.27 | 0.98 | 0.77 | 0.54 | 0.76 |
| | 71 | 10×7 | 112 | 0.01 | 0.13 | 0.74 | 0.95 | 0.60 | -0.10 | 0.63 | 0.71 | 0.41 | 0.99 | 0.79 | 0.64 | 0.80 |
| | 211 | 30×7 | 112 | 0.01 | 0.64 | 0.81 | 0.98 | **0.81** | 0.00 | 0.80 | 0.97 | **0.59** | 1.00 | 0.81 | 0.68 | 0.83 |
| | 351 | 50×7 | 112 | 0.01 | 0.66 | 0.76 | 0.97 | 0.80 | 0.20 | 0.37 | 0.93 | 0.50 | 1.01 | 0.82 | 0.73 | **0.85** |

evaluation loss instead of proxies.

## 3.3. Evaluation Metrics

We conduct two kinds of experiments to evaluate DeMix: Proxy Consistency and Mixture Quality. Proxy Consistency consists of two metrics: proxy accuracy for validating the ranking consistency of the proxy against reference models, and capability recovery indicates how effectively do proxy models maintain absolute performance. While Mixture Quality assesses the downstream performance of the model trained on final data mixture through benchmark score and rank.

### 3.3.1. PROXY CONSISTENCY

**Proxy Accuracy** Proxy Accuracy is defined as the measure of ranking consistency between the proxy models and the ground-truth reference models. To evaluate this, we randomly sample 96 mixture ratios and train 96 corresponding reference models on a large-scale corpus of 50B tokens, serving as the performance standard. In parallel, proxy models are constructed via efficient model merging. We calculate the Spearman's rank correlation coefficient $\rho$ (Spearman, 1961) between the benchmark scores of the proxies and references to quantify their alignment. Additionally, to verify the precision in identifying high-performing mixtures, we report the Spearman's $\rho$ specifically for the top 25% reference models. A higher $\rho$ indicates that the proxy models can accurately predict the relative performance of data mixtures, validating the effectiveness of our training-free approach.

**Capability Recovery** To quantitatively assess the extent of performance retention in the merged models, we introduce the Capability Recovery Rate. This metric is defined as the quotient of the average benchmark score of the proxy

model to that of the corresponding reference model. By directly comparing absolute performance levels, the Capability Recovery Rate serves as a critical indicator of how effectively the proxy model inherits and upholds the fundamental competencies of the reference model without incurring additional training costs.

### 3.3.2. MIXTURE QUALITY

**Benchmark Score and Rank** We evaluate the final mixture ratio by training a model on 50B tokens with this mixture and evaluating it on general, math, and code benchmarks, aiming for versatility. Since the benchmark scores vary substantially across different domains, we report relative rankings with respect to the 96 reference models, and use the macro-averaged rank across general language understanding, mathematical reasoning, and code generation benchmarks as the final rank metric.

## 4. Experimental Results

### 4.1. Proxy Consistency

Table 2 compares the cost-effectiveness and proxy accuracy of DeMix against conventional training-based approaches. We fix the number of proxies to 112, aligning with the configuration used in CLIMB. Accordingly, we arrive at the following conclusion:

**DeMix provides accurate proxies at significantly lower cost.** The results demonstrate that DeMix produces sufficiently strong and accurate proxies, significantly outperforming training-based baselines under the same computational budget. Specifically, when merging component models trained on 30B tokens, DeMix achieves a macro avg. $\rho$ of 0.81 and top-25% $\rho$ of 0.59, while consuming a total token budget of only 212B. In contrast, the training-based approach reaches only 0.53 and 0.20 under a comparable

*Table 3.* Comparison of the optimal mixture performance obtained from different data mixing methods. † is the benchmarking cost derived from equal GPU hour. The best average rank score across all methods is marked in bold, and the best score for the optimal setting within each method is underlined.

| Method | Train Cost (B) ↓ | | | | Benchmarks (%) ↑ | | | | | | | | | | | | | Rank ↓ | | | |
| | Total Budget | Pre-Cost | No. Proxies | Budget / Proxy | General | | | | | | Code | | | Math | | | General | Code | Math | Macro Avg. |
| | | | | | ARC-E | HellaSwag | PIQA | SIQA | WinoGrande | Avg. | MBPP | HumanEval | Avg. | GSM8K | MATH | Avg. | | | | |
|---|---|---|---|---|---|---|---|---|---|---|---|---|---|---|---|---|---|---|---|---|
| Uniform | - | - | - | - | 71.34 | 54.33 | 73.18 | 40.52 | 55.67 | 59.01 | 18.50 | 18.19 | 18.34 | 12.50 | 6.74 | 9.62 | 9 | 44 | 57 | 36.67 |
| Heuristic | - | - | - | - | 69.57 | 53.72 | 71.75 | 40.32 | 55.27 | 58.13 | 24.27 | 24.90 | 24.58 | 18.20 | 10.16 | 14.18 | 94 | 5 | 28 | 42.33 |
| RegMix | 224 | - | 112 | 2 | 70.81 | 54.34 | 73.45 | 40.41 | 54.58 | 58.72 | 17.70 | 20.58 | 19.14 | 19.13 | 10.43 | 14.78 | 52 | 41 | 21 | 38.00 |
| | 224 | - | 28 | 8 | 70.58 | 54.62 | 73.21 | 40.36 | 55.77 | 58.91 | 16.92 | 23.78 | 20.35 | 14.89 | 7.91 | 11.40 | 22 | 29 | 54 | 35.00 |
| | 448 | - | 56 | 8 | 71.43 | 54.55 | 73.54 | 40.76 | 55.63 | 59.18 | 18.33 | 21.85 | 20.09 | 15.30 | 7.96 | 11.63 | 2 | 32 | 50 | 28.00 |
| CLIMB | 224 | - | 112 | 2 | 70.75 | 54.44 | 73.41 | 40.37 | 54.93 | 58.78 | 17.47 | 20.32 | 18.90 | 19.21 | 10.65 | 14.93 | 41 | 42 | 21 | 34.67 |
| | 224 | - | 28 | 8 | 71.63 | 54.79 | 73.51 | 40.56 | 55.50 | 59.20 | 15.83 | 24.19 | 20.01 | 13.67 | 7.36 | 10.52 | 2 | 32 | 56 | 30.00 |
| | 448 | - | 56 | 8 | 70.31 | 54.19 | 73.22 | 40.40 | 55.57 | 58.74 | 19.53 | 22.66 | 21.10 | 20.53 | 11.61 | 16.07 | 47 | 25 | 11 | 27.67 |
| DeMix (Ours) | 211 | 30×7 | 56 | 0.01† | 70.37 | 53.43 | 73.21 | 40.52 | 55.29 | 58.56 | 21.63 | 21.95 | 21.79 | 24.43 | 14.10 | 19.26 | 66 | 21 | 1 | 29.33 |
| | 211 | 30×7 | 112 | 0.01 | 70.81 | 53.71 | 73.26 | 40.34 | 55.96 | 58.81 | 19.73 | 22.56 | 21.15 | 19.75 | 10.13 | 14.94 | 31 | 25 | 21 | 25.67 |
| | 212 | 30×7 | 224 | 0.01 | 71.08 | 53.78 | 72.94 | 40.63 | 55.43 | 58.77 | 20.70 | 24.29 | 22.49 | 20.98 | 10.55 | 15.76 | 46 | 12 | 14 | **24.00** |
| | 214 | 30×7 | 448 | 0.01 | 70.61 | 54.29 | 73.12 | 40.22 | 55.89 | 58.83 | 22.03 | 23.17 | 22.60 | 15.73 | 8.35 | 12.04 | 31 | 10 | 42 | 27.67 |

budget. To attain a similar performance level, it requires a prohibitive 1344B tokens, corresponding to a 6.4× increase in cost relative to our method. Additionally, DeMix maintains a high capability recovery rate (up to 0.85), confirming that weighted model merging serves as a highly reliable and efficient proxy for real data mixtures. Based on these results, we select component models trained with 30B tokens for subsequent experiments due to their efficiency and effectiveness.

## 4.2. Mixture Quality

We present the performance of data mixtures produced by different methods in Table 3, with specific mixture details provided in Appendix C. For the baseline methods RegMix and CLIMB, we evaluate their performance using both 2B and 8B proxy models across a varying number of proxies. We also report results for Uniform and Heuristic strategies to provide essential comparisons against other tuned baselines. The results demonstrate the following:

**DeMix achieves superior mixture quality with lower training cost.** The total training budget of DeMix remains roughly unchanged as the number of proxies increases. Among all baseline methods, DeMix with 224 merged proxies (the green row) achieves the top performance rank of 24.00. Under a comparable training budget, neither RegMix nor CLIMB delivers comparable results. Specifically, neither 112 2B-trained proxies nor 28 8B-trained proxies surpass DeMix. Moreover, 2B-trained proxies yield inferior performance relative to 8B-trained proxies under the same budget, suggesting small proxy performs poorly for this type

of task. When the training budget is scaled up to 448B with 56 8B-trained proxies, both RegMix and CLIMB exhibit performance improvements. Yet DeMix still outperforms these methods with a substantially lower training budget.

**Scaling the proxy count enhances the mixture quality within a certain range.** For DeMix, the performance shows a continuous upward trend as the proxy count is scaled from 56 to 224, with the rank improving from 29.33 to 24.00. A similar improvement is also observed for the 8B-trained proxy versions of RegMix and CLIMB when the proxy count increases from 28 to 56. Further performance gains can be reasonably anticipated, yet such gains would be accompanied by substantial computational overhead. This phenomenon illustrates that the proxy count plays a critical role in determining mixture quality. Meanwhile, the rank of DeMix is observed to decline once the proxy count reaches 448, indicating that an excessively large number of proxies may introduce the risk of overfitting noise.

Nevertheless, it is of great importance to explore a broader range of proxies. For instance, expanding the pool of candidate datasets will increase the dimensionality of the mixture ratios, which in turn increases the number of proxies required to conduct an effective search. Therefore, DeMix retains a distinct advantage in reducing the computational burden associated with the search procedure.

## 4.3. Ablation Study

We conduct ablation and validation studies to identify the key factors that affect DeMix and to examine its robustness

*Table 4.* Comparison of different merging methods. HP-Free denotes hyperparameter-free. Linear merging serves as a simple and effective method.

| Method | HP-Free | $\rho$ | Capability Recovery |
|---|---|---|---|
| **Linear** | ✓ | **0.787** | **0.845** |
| Multi-SLERP | ✓ | 0.785 | 0.813 |
| Breadcrumbs | ✓ | 0.735 | 0.831 |
| DARE | ✗ | 0.757 | 0.835 |
| DELLA | ✗ | 0.784 | 0.778 |
| TIES | ✗ | 0.783 | 0.786 |

*Table 5.* Comparison of performances with different proportion of mixed general data in candidate dataset.

| General Data Proportion | $\rho$ | Capability Recovery |
|---|---|---|
| **50%** | **0.787** | **0.845** |
| 25% | 0.667 | 0.796 |
| 0% | 0.652 | 0.795 |

across settings. We first ablate the merging strategies and the proportion of general data in the candidate datasets. For these experiments, we report the mean macro-average $\rho$ and capability recovery rate across 96 different mixture ratios, using component models trained on 30B, 40B, and 50B tokens. We further evaluate the optimized mixture ratios on Qwen3-8B and study a finer-grained partition of the training data, aiming to test whether DeMix transfers to larger-scale models and remains reliable under different partition granularities.

### 4.3.1. MERGING METHODS

We compare different merging methods by evaluating the accuracy of their corresponding proxies in Table 4. Results are shown for Linear (Wortsman et al., 2022a), Multi-SLERP (Goddard et al., 2024), DARE (Yu et al., 2024), Breadcrumbs (Davari & Belilovsky, 2024), DELLA (Deep et al., 2024), and TIES (Yadav et al., 2023). Among others, Linear is an intuitive, simple, and hyperparameter-free (HP-free) method that achieves the best capability recovery rate and macro-average $\rho$.

### 4.3.2. PROPORTION OF CANDIDATE DATASETS

Prior to training the component model, we incorporate general data into the candidate data at a specific ratio. Table 5 presents the impact of different mixing ratios on the final proxy accuracy. When the ratio is reduced to 25%, the proxy accuracy drops significantly to a $\rho$ value of 0.667, with a capacity recovery rate of 0.796. The performance further declines when the ratio decreases to 0%. These results illustrate that the absence or insufficient proportion of general

data results in a sharp reduction in both $\rho$ and the capacity recovery rate. This validates the necessity of employing this data regularization technique.

### 4.3.3. VALIDITY ON LARGER SCALE MODELS

*Table 6.* Comparison of the optimal mixture performance on Qwen3-8B obtained from different data mixing methods

| Method | Proxy | Avg. ↑ | | | Rank ↓ | | | |
|---|---|---|---|---|---|---|---|---|
| | | Gen. | Code | Math | Gen. | Code | Math | Avg. |
| Uniform | - | 62.07 | 24.60 | 18.52 | 7 | 54 | 64 | 41.67 |
| Heuristic | - | 59.86 | 27.87 | 25.58 | 64 | 18 | 37 | 39.67 |
| RegMix | 112×2 | 61.49 | 24.44 | 27.16 | 30 | 56 | 31 | 39.00 |
| | 28×8 | 61.85 | 24.62 | 22.30 | 15 | 53 | 56 | 41.33 |
| | 56×8 | 62.02 | 25.77 | 23.52 | 10 | 41 | 51 | 34.00 |
| CLIMB | 112×2 | 60.82 | 23.98 | 29.80 | 51 | 57 | 17 | 41.67 |
| | 28×8 | 62.10 | 25.10 | 24.50 | 7 | 49 | 46 | 34.00 |
| | 56×8 | 61.78 | 26.96 | 24.25 | 19 | 25 | 46 | 30.00 |
| DeMix (Ours) | 56 | 60.32 | 27.61 | 34.14 | 63 | 21 | 10 | 31.33 |
| | 112 | 60.81 | 27.80 | 28.94 | 51 | 18 | 20 | 29.67 |
| | 224 | 60.83 | 27.36 | 29.68 | 49 | 21 | 17 | 29.00 |
| | 448 | 61.19 | 30.60 | 25.02 | 36 | 4 | 43 | **27.67** |

To further evaluate the scalability of DeMix, we conduct experiments on Qwen3-8B using the same mixture ratios as those reported in Table 3. As shown in Table 6, the mixture ratios produced by DeMix achieve a better average rank compared with alternative ratios. This observation is consistent with the results on Qwen3-1.7B, suggesting that the effectiveness of DeMix is not limited to smaller models and can transfer to larger-scale models.

### 4.3.4. FINER PARTITIONING OF TRAINING DATA

We further examine a finer data partition with 15 categories, compared to the 7-category partition used in the main experiments in Table 2. The overall findings remain consistent. As shown in Table 7, the proxy accuracy of DeMix under the finer partition follows the same trend as that reported in Table 2. In particular, DeMix with 30B/50B component models achieves Spearman correlations comparable to those of the 12B-trained proxy, with significantly lower training costs. This suggests that the proxy remains reliable under finer partitions and that the main conclusions are not sensitive to the choice of partition granularity.

Nevertheless, finer partitioning also comes with important trade-offs. While it enlarges the exploration space and may potentially yield better mixing ratios, the search space grows rapidly as the number of categories increases. Compared with coarser partitions, finer partitions benefit less from the prior information provided by merging similar datasets, and therefore may perform worse under the same search budget. They also incur higher computational cost, requiring more component models in DeMix or more proxy training runs in methods such as RegMix and CLIMB. Moreover, finer parti-

*Table 7.* Comparison of training costs and proxy accuracy between DeMix and training-based proxies with finer 15 dataset partitions.

| Method | Train Cost (B) ↓ | | | | Spearman's ρ ↑ | | | | Top 25% Spearman's ρ ↑ | | | | Capability Recovery ↑ | | | |
| | Total Budget | Pre-Cost | No. Proxies | Budget / Proxy | General | Code | Math | Macro Avg. | General | Code | Math | Macro Avg. | General | Code | Math | Macro Avg. |
|---|---|---|---|---|---|---|---|---|---|---|---|---|---|---|---|---|
| Trained Proxy (RegMix/CLIMB) | 224 | - | 112 | 2 | 0.61 | 0.46 | 0.92 | 0.66 | 0.23 | 0.74 | 0.26 | 0.41 | 0.98 | 0.81 | 0.53 | 0.77 |
| | 448 | - | 112 | 4 | 0.66 | 0.82 | 0.95 | 0.81 | 0.00 | 0.74 | 0.49 | 0.41 | 0.98 | 0.83 | 0.61 | 0.81 |
| | 896 | - | 112 | 8 | 0.66 | 0.84 | 0.94 | 0.81 | 0.31 | 0.63 | 0.51 | 0.48 | 0.99 | 0.86 | 0.68 | 0.84 |
| | 1344 | - | 112 | 12 | 0.63 | 0.94 | 0.96 | **0.84** | 0.17 | 0.89 | 0.69 | **0.58** | 0.99 | 0.88 | 0.74 | **0.87** |
| DeMix (Ours) | 31 | 2×15 | 112 | 0.01† | -0.08 | 0.49 | 0.93 | 0.45 | -0.03 | 0.46 | 0.36 | 0.26 | 0.98 | 0.81 | 0.55 | 0.78 |
| | 151 | 10×15 | 112 | 0.01 | 0.69 | 0.79 | 0.91 | 0.80 | -0.03 | 0.89 | 0.50 | 0.45 | 0.99 | 0.82 | 0.67 | 0.83 |
| | 451 | 30×15 | 112 | 0.01 | 0.73 | 0.82 | 0.93 | 0.83 | 0.29 | 0.91 | 0.56 | **0.59** | 1.00 | 0.86 | 0.71 | 0.86 |
| | 751 | 50×15 | 112 | 0.01 | 0.70 | 0.84 | 0.97 | **0.84** | 0.34 | 0.91 | 0.44 | 0.56 | 1.01 | 0.86 | 0.73 | **0.87** |

tions often result in many categories with very small mixing proportions, for which it is difficult to reliably estimate how changes in their proportions affect final model performance. As a result, the optimized ratios of such small categories may be less meaningful. For these reasons, practical pre-training recipe design typically avoids overly fine-grained partitions (Feng et al., 2024; Blakeman et al., 2025).

# 5. DeMix Corpora

*Table 8.* Comparison of public high-quality pre-training datasets.

| Dataset | Multilingual | Math& Code | Validated Mixture |
|---|---|---|---|
| DCLM-baseline | ✗ | ✗ | ✗ |
| FineWeb-Edu | ✗ | ✗ | ✗ |
| ClimbMix | ✗ | ✗ | ✗ |
| DOLMA-v1.7 | ✗ | ✓ | ✗ |
| SmolLM-Corpus | ✓ | ✓ | ✗ |
| Nemotron-Pretrain | ✓ | ✓ | ✗ |
| **DeMix Corpora** | ✓ | ✓ | ✓ |

Public pre-training corpora can be categorized into several types. Web-derived general English corpora include FineWeb-Edu (Penedo et al., 2024), DCLM-baseline (Li et al., 2024), and ClimbMix (Diao et al., 2025); composite corpora include SmolLM-Corpus (Ben Allal et al., 2024), DOLMA (Soldaini et al., 2024), and Nemotron-Pretrain (Basant et al., 2025). In addition, multilingual (Messmer et al., 2025; De Gibert et al., 2024), code (Li et al., 2023), and mathematics (Allal et al., 2025a) corpora are not directly applicable for pre-training. As shown in Table 8, there remains a scarcity of high-quality corpora with validated mixture. In contrast, our proposed DeMix Corpora (15T original tokens and 22T mixture tokens) serves as a comprehensive, high-quality, large-scale, and carefully mixed resource that can be directly employed for pre-training.

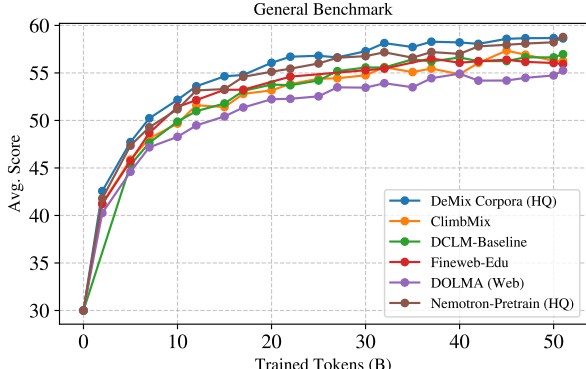

*Figure 3.* General performance of high-quality general datasets.

*Table 9.* Multi-Domain performance of mixed datasets after mid-training 50B tokens.

| Dataset | General | Code | Math | Avg. Rank↓ |
|---|---|---|---|---|
| SmolLM-Corpus | 59.13 | 21.01 | 9.14 | 31.33 |
| Nemotron-Pretrain | 57.67 | 28.35 | 16.12 | 36.00 |
| **DeMix Corpora** | 58.77 | 22.49 | 15.76 | **24.00** |

To ensure high data quality, we curate the corpora from heterogeneous open-source sources, followed by a comprehensive data cleaning pipeline (see Appendix A). As shown in Figure 3, when training a Qwen3-1.7B model from scratch and evaluating on general benchmarks, the high-quality (HQ) general data subset of the DeMix Corpora outperforms all other general datasets.

In addition, DeMix Corpora not only features large and high-quality general corpus subset, but also adopts a validated optimal data mixture that best balances multi-domain capabilities of pre-training. The final mixtures are illustrated in Figure 4. We compare DeMix Corpora with existing mixed

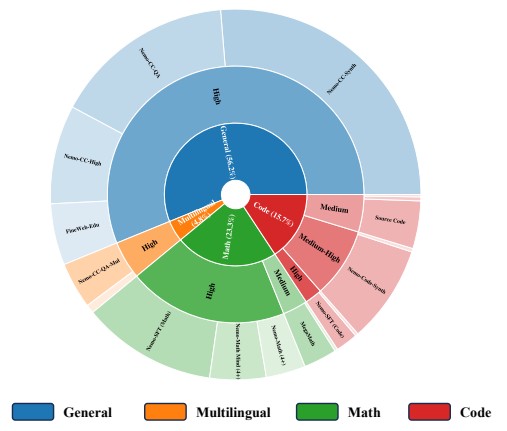

*Figure 4.* The data mixture constructed using our DeMix framework. The three hierarchical levels from the inside out are domain, data category, and data origin.

datasets. The results presented in Table 9 demonstrate that DeMix Corpora achieves a superior balance between general and domain-specific capabilities, yielding the best average rank of 24.00. In contrast, Nemotron-Pretrain exhibits poor general capability due to an insufficient volume of general-domain training data. SmolLM-Corpus features high-quality general data with a large proportion, but its math capability is severely deficient.

## 6. Related Works

### 6.1. Data Mixture

Data mixture plays a critical role in successful LLM pre-training (Chen et al., 2023; Shen et al., 2023; Ye et al., 2024; Xie et al., 2023). Many LLM teams conduct large-scale proxy experiments using mid-sized models with a sufficient token budget (Li et al., 2025; Nie et al., 2025; Blakeman et al., 2025). While such approaches can yield accurate estimates given massive computational resources, their cost makes them impractical for many research institutes. Recently, researchers have begun to propose automated methods for data mixture optimization. RegMix (Liu et al., 2024) samples a large number of data ratios for tiny-scale proxy experiments and generates optimal predictions based on the results of a regression predictor. CLIMB (Diao et al., 2025) iteratively samples and calibrates sample points with higher predicted scores, thereby improving the prediction accuracy for high-performing samples. However, such automated methods are only validated to optimize simple general capabilities. When jointly optimizing more challenging tasks such as math and code (Kimi Team et al., 2025; Feng et al., 2024; Basant et al., 2025), tiny-scale proxy experiments tend to become inaccurate due to insufficient training (Allal et al., 2025b; Li et al., 2025).

### 6.2. Model Merging

Model merging has emerged as a promising training-free approach that combines multiple LLMs with identical structures into a new LLM using a series of arithmetic operations. (Goddard et al., 2024; Wortsman et al., 2022a; Ilharco et al., 2022; Yadav et al., 2023; Davari & Belilovsky, 2024). This technique can be used to regularize models (Luo et al., 2025; Wortsman et al., 2022b), improve generalization (Izmailov et al., 2018; Yang et al., 2024), mitigate catastrophic forgetting (Xiao et al., 2024), or even replace fine-tuning entirely (Ahmadian et al., 2024). Recent works have been exploring model merging in data selection. For instance, Merge-to-Mix (Tao et al., 2025) employs unweighted model averaging to enumerate all binary subset choices to discard datasets during fine-tuning. In contrast, optimizing pre-training data mixtures is substantially more challenging: mixing weights are continuous-valued and the feasible space is unbounded. Recently, several studies have shown that merging weight deltas from models sharing the same base but trained on different datasets is a highly effective alternative to directly merging their training processes, provided that parameter updates remain relatively small (Wu et al., 2025; Lin et al., 2025b). The concurrent work MergeMix (Wang et al., 2026) uses a similar approach for mid-training. Our work significantly extend this line of work by leveraging weighted model merging as a proxy for real data mixture during pre-training. By ensuring that a single proxy is adequately accurate, we eliminate the overhead of training, thereby enabling extensive sampling to search for the optimal ratio.

## 7. Conclusion

In this work, we introduce DeMix, a pre-training data mixture optimization framework that decouples mixture search from costly proxy training by leveraging weighted model merging. DeMix simultaneously achieves sufficiency (unlimited proxy models), accuracy (faithful proxy performances) and efficiency (fixed token budgets). Across comprehensive evaluations, DeMix yields the best data mixture that balances the diverse capability demands of LLM pre-training across general language understanding, mathematical reasoning, and code generation. To further support reproducible research and practical pre-training, we release DeMix Corpora, a 22T-token high-quality dataset accompanied by validated mixture ratios, providing a resource for large-scale LLM pre-training development.

## Acknowledgements

We thank the reviewers for their positive feedback and constructive comments, which helped improve the quality of this paper. We are also grateful to the co-authors who contributed to the development and maintenance of the dataset

construction pipeline, which provided important support for this work.

## Impact Statement

This paper presents research aimed at advancing the development of large language models. The proposed method is designed to facilitate data mixture optimization for large language model pre-training. The dataset in this work is constructed by filtering and merging multiple datasets held under valid and appropriate licenses, with no associated legal or ethical risks identified.

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

# A. Details of DeMix Corpora

## A.1. Data Curation Pipeline

This subsection presents a unified pipeline for building the DeMix Corpora from heterogeneous sources including general-domain, multilingual, mathematical, and code data. Global exact and fuzzy deduplication is scaled to enhance the corpus's overall quality and eliminate redundant content; dataset-specific perplexity filtering via a lightweight scoring model further refines data validity and relevance. A FastText-based quality classifier, trained on well-constructed positive-negative data mixtures and validated through controlled pre-training, is leveraged to identify and retain semantically meaningful high-quality data samples. In addition, for Chinese general-domain corpora, we introduce a dedicated quality classifier to filter low-quality samples and upsample high-quality signals, further improving the quality distribution of the Chinese data. Finally, hierarchical instance-level labeling—bootstrapped from high-confidence large-model annotations and distilled into a lightweight labeler—is introduced to enable subsequent data analysis and targeted data selection by category, with the corpus quality additionally verified through small-model training experiments and benchmark comparisons.

**Data Collection**    The general-domain data we collected mainly covers well-known open-source datasets such as FineWeb-Edu (Penedo et al., 2024), DCLM-Baseline (Li et al., 2024), DOLMA-v1.7(Soldaini et al., 2024), Ultra-FineWeb (Wang et al., 2025), and the NVIDIA Nemotron corpus (Basant et al., 2025). For mathematical data, we include sources such as FineMath (Liu et al., 2025), MegaMath (Zhou et al., 2025), and SwallowMath (Fujii et al., 2025). For code data, we include sources such as OpenCoder (Huang et al., 2025) and SwallowCode (Fujii et al., 2025). In addition, we incorporated multilingual data and reasoning data. A complete list of our data sources is provided in Table 10.

**Deduplication**    We perform global exact deduplication and fuzzy deduplication on most datasets (Lee et al., 2022), while retaining the original data only for confirmed high-quality small-scale datasets. Although some works (Penedo et al., 2024; 2025) argue that global deduplication may remove a substantial number of high-quality samples, we still adopt it for large-scale web data, primarily because the greatly increased data volume mitigates the impact of this issue. For fuzzy deduplication, we extract 24-grams from each document, compute 260 MinHash functions per document, and partition them into 20 bands of 13 hashes each, targeting documents with at least 90% similarity. Any pair of documents that share an identical 13-MinHash signature in any band is considered duplicates.

**Perplexity Filtering**    We perform perplexity-based data filtering using the Qwen3-0.6B base model as the scoring model. For each dataset, we manually inspect high-perplexity samples and determine dataset-specific thresholds. Based on these thresholds, samples with extremely low perplexity are removed, resulting in an overall data reduction of approximately 2%.

**FastText Filtering**    We construct a binary data quality classifier based on FastText. Both the positive and negative sets contain over one million samples. The positive set is composed of randomly sampled data, low-perplexity data, and high-quality subsets from ELI5-category (Fan et al., 2019) and OpenHermes-2.5 (Teknium, 2023), while the negative set is partially sourced from Falcon-RefinedWeb (Penedo et al., 2023) and high-perplexity data. Since high-perplexity data is dominated by short samples, we apply weighted sampling to long samples to increase the likelihood of selecting low-quality long-form data. To determine appropriate source proportions for the positive and negative sets, we construct multiple candidate sample mixtures and train Qwen3-0.6B from scratch on each set separately. Experimental results show that the model trained on the positive set achieves a lower average evaluation loss than the model trained on the negative set, demonstrating a clear quality distinction between the two sets. Based on these validated samples, we subsequently train a FastText classifier. The resulting classifier removes approximately 3% of the English corpus and effectively identifies low-quality web and code data.

**Quality Filtering for Chinese General-Domain Corpora**    We propose a quality assessment and sampling framework for large-scale Chinese general-domain web corpora, inspired by prior work such as FineWeb-Edu(Penedo et al., 2024). In this framework, education-related and information-dense signals are treated as proxies for high-quality text, with the goal of preferentially retaining and emphasizing content with higher informational value and structural coherence during corpus construction. Specifically, we categorize text samples into five quality levels: undetermined (e.g., non-Chinese or nonsensical content), extremely low quality, low quality, high quality, and extremely high quality. The labeling criteria jointly consider linguistic completeness, information density, structural coherence, and the degree to which the text exhibits knowledge-bearing or explanatory characteristics, without restricting the data to explicitly educational contexts. To obtain reliable quality annotations, we first manually labeled 5,000 Chinese samples and fine-tuned a 32B-parameter language

model to learn the above quality distinctions. We then used this model to automatically annotate approximately 1 million Chinese web documents. Based on these pseudo-labeled samples, we trained a lightweight text quality classifier, which uses gte-multilingual-base as the text embedding backbone followed by a classification head. During corpus construction, the trained classifier is applied to remove samples classified as extremely low quality or undetermined, while samples labeled as extremely high quality are upsampled, thereby improving the overall quality distribution and structural characteristics of Chinese general-domain training data.

**Instance-Level Data Labeling**   To estimate the domain distribution of our pre-training corpus, we build a three-level hierarchical taxonomy. We define level-1 labels following the standard graduate disciplinary classification, and derive level-2 and level-3 labels under each parent via Gemini 2.5 Pro (Comanici et al., 2025) and GPT-4o (Achiam et al., 2023). We then annotate the corpus using Qwen3-235B-A22B (Yang et al., 2025) and retain 3.63M high-confidence labeled instances to train a 4B model, which is used to label the full pre-training dataset.

**Data Evaluation**   To validate the quality of individual datasets, we train a relatively small model (Qwen-3-1.7B) on 50B tokens. For general datasets, we train from scratch. For math/code datasets, we mix in 80%/60% general data and train from a pre-trained mid checkpoint. The benchmarks we use are widely adopted, ensuring that they can exceed random performance after a small amount of training and maintain stable ranking across subsequent training stages.

**Candidate Data Preparation**   DeMix is leveraged in the late stages of pre-training: for general data, we select only the highest-quality tier; for math and code, we discard the lowest-quality tiers, stratify the remaining corpora by both category and quality, and merge similar sources to reduce the dataset count. This constitutes a common engineering trade-off (Diao et al., 2025) and a standard practice for cross-domain data mixing (Basant et al., 2025; Feng et al., 2024). By reducing the dimensionality of mixture ratios, we substantially alleviate the search burden of identifying a high-performance mixture. While this may lower the attainable performance upper bound, it is a necessary compromise given the exorbitant cost of large-scale pre-training. Ultimately, we obtain seven dataset categories that require fine-grained mixing. For each candidate dataset, we mix it with 50% general data as a form of regularization, since an excessive proportion of domain-specific data can significantly degrade the model's general capabilities during LLM pre-training (Lin et al., 2025a; Bansal & Sanghavi, 2025; Allal et al., 2025b). Although we can only search within the subspace where non-general data accounts for less than 50%, this constraint—that non-general data should not dominate the pre-training corpus—is consistent with the consensus (Basant et al., 2025; Blakeman et al., 2025; Allal et al., 2025a).

**Component Model Training**   While DeMix is adopted for the late stages, for early pre-training, the model exhibits limited capacity and thus cannot tackle more challenging tasks; we therefore evaluate data mixtures using general benchmarks, which is a single-objective optimization goal that can be cultivated during early training, making the construction of data mixtures relatively simple and straightforward. In practice, we find that direct upsampling/downsampling based on quality scores yield consistent and rational data mixtures. We train a 1.7B model from scratch using the stage-1 mixture for a sufficient number of tokens, which serves as the base model for component models. We then train seven component models $\{M_i\}$ on each candidate dataset $\{D_i\}$ on 50B tokens, ensuring that they possess sufficient emergent capabilities to solve challenging math and code problems.

### A.2. Data Composition Across Stages

The data mixture used in LLM pre-training is inherently dynamic rather than static (Feng et al., 2024; Basant et al., 2025; Blakeman et al., 2025). During the initial phases, the priority is typically given to data diversity, whereas the later stages shift focus toward high data quality (Wang et al., 2021). In practice, early training stages typically emphasize data diversity, while late stages increasingly prioritize high-quality data to refine advanced capabilities (Wang et al., 2021). Consequently, pre-training is often organized into multiple stages, with the proportion of high-quality math and code data significantly increased in the late stages (Basant et al., 2025; Allal et al., 2025a). As a result, modern pre-training pipelines are often organized into multiple stages, with the proportion of high-quality math and code data substantially increased in the late stages (Basant et al., 2025; Allal et al., 2025a).

In practice, we partition the pre-training data into three stages, with the proportion of high-quality data such as mathematical and coding data gradually increasing across successive stages. In the first stage, as the model primarily acquires broad general knowledge during the initial phase of training, data quality exerts a less significant impact than in subsequent stages. Consequently, we only adjust the data mixture using basic general benchmarks, performing upsampling and downsampling

*Table 10.* Composition of DeMix Corpora and the mixture in each stage, measured in tokens (B).

| Category | Quality | Original Source | Amount | Stage 1 | Stage 2 | Stage 3 |
|---|---|---|---|---|---|---|
| General | High | FineWeb-Edu | 199 | 740 | 369 | 107 |
| | | Nemo-CC-High | 385 | 1191 | 594 | 172 |
| | | Nemo-CC-Synth | 1106 | 3630 | 1812 | 525 |
| | | Nemo-CC-QA | 546 | 2190 | 1093 | 317 |
| | Medium-High | DCLM | 505 | 505 | 0 | 0 |
| | | DOLMA (Social) | 71 | 71 | 0 | 0 |
| | | Nemo-CC-Med-High | 393 | 393 | 0 | 0 |
| | | OpenCoder (Web) | 38 | 38 | 0 | 0 |
| | | Ultra-FineWeb (en) | 79 | 79 | 0 | 0 |
| | | DOLMA (Others) | 54 | 54 | 0 | 0 |
| | | Nemo-SFT (General) | 67 | 67 | 0 | 0 |
| | Medium | DOLMA (Web) | 1278 | 774 | 0 | 0 |
| | | Web Crawl (en) | 1417 | 330 | 0 | 0 |
| | | Nemo-CC-Medium | 1605 | 965 | 0 | 0 |
| | | DOLMA (Scholar) | 77 | 77 | 0 | 0 |
| Multilingual | High | Ultra-FineWeb (zh) | 102 | 102 | 51 | 15 |
| | | Nemo-CC-QA-Mul | 562 | 562 | 280 | 81 |
| | Medium | Web Crawl (zh) | 3304 | 991 | 0 | 0 |
| Math | High | Nemo-Math Mind (4+) | 69 | 69 | 199 | 98 |
| | | Nemo-Math (4+) | 49 | 49 | 141 | 69 |
| | | Nemo-SFT (Math) | 166 | 166 | 479 | 235 |
| | Medium-High | Reason (Math) | 38 | 38 | 36 | 0 |
| | | SwallowMath | 31 | 31 | 29 | 0 |
| | Medium | MegaMath | 92 | 92 | 87 | 58 |
| | | FineMath (4+) | 8 | 8 | 7 | 5 |
| | Low | Nemo-Math (3) | 76 | 76 | 0 | 0 |
| | | FineMath (3) | 20 | 20 | 0 | 0 |
| | | Math (Others) | 48 | 48 | 0 | 0 |
| Code | High | Nemo-SFT (Code) | 45 | 45 | 150 | 39 |
| | | OpenCoder | 6 | 6 | 20 | 5 |
| | Medium-High | Nemo-Code-Synth | 136 | 136 | 410 | 170 |
| | | MegaMath (Code) | 5 | 5 | 15 | 6 |
| | Medium | Source Code | 559 | 559 | 200 | 83 |
| | | SwallowCode | 47 | 47 | 17 | 7 |
| | | Reason (Code) | 27 | 27 | 10 | 4 |
| | Low | StarCoder | 207 | 207 | 0 | 0 |
| **Total** | | | **13417** | **14388** | **5999** | **1996** |

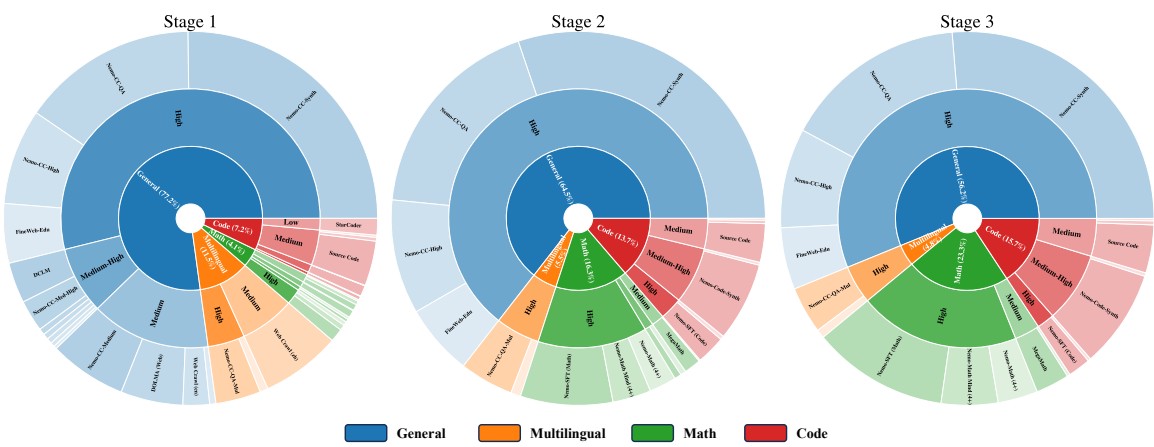

*Figure 5.* Data mixtures for the three stages of pre-training in the DeMix Corpora, with approximately 14T, 6T, and 2T tokens, respectively. The three hierarchical levels from the inside out are domain, data category, and data origin.

based on quality scores. In contrast, during Stages 2 and 3, data quality directly influences the final training performance; thus, we employ the DeMix framework to optimize the data mixture ratio. Furthermore, to prevent excessive repetition of individual samples, we reduce the threshold for the allowable number of repetitions in Stage 2, yielding a more balanced data distribution compared to Stage 3.

We presents the detailed composition of DeMix Corpora in Figure 5 and Table 10, including the amount of tokens after our curation pipeline and the token allocation in 3 training stages after data mixing by DeMix.

## B. Baseline Details.

For RegMix and CLIMB, the only differences from DeMix lie in how proxy models are obtained and how the predictor is trained. For RegMix, we sample a required number of mixtures and train the model from the same initialization as the DeMix component models, using a fixed number of tokens. We then evaluate the trained models on to obtain benchmark scores, and fit a LightGBM predictor using the same hyperparameters as in DeMix. Finally, we apply the trained predictor to the same large set of sampled candidates and select the top-ranked mixtures as the final optimal mixture. For CLIMB, the only difference is that it adopts iterative sampling. Consistent with our DeMix procedure, it performs three iterations: for 28 points, we sample $16 + 8 + 4$; for 56 points, $32 + 16 + 8$; for 112 points, $64 + 32 + 16$.

## C. Detailed Mixtures in Experiments.

We extend Table 3 by listing the detailed mixture ratios in Table 11.

## D. Additional Verification of the Model-Merging Approximation

The approximation in Eq. 5 is mainly motivated by prior studies on model merging and small-update regimes. To further provide empirical evidence for its validity in our setting, we compare the merged proxy model with a model trained directly on mixed data. The goal is to verify whether the merged proxy model can consistently approximate the performance of the corresponding mixed-data-trained model, especially when the magnitude of parameter updates is relatively small.

Specifically, we select one math dataset as $D_1$ and one code dataset as $D_2$, and consider several mixture proportions between them. For each mixture proportion, we compare two types of proxy models:

- **Model merge**: we first train two component models separately on $D_1$ and $D_2$, and then merge their parameters according to the target mixture ratio.

- **Data mix**: we directly train a model on the mixed dataset $D_1 + D_2$ with the same target mixture ratio.

*Table 11.* Detailed mixtures from different experiments.

| Method | Train Cost (B) ↓ | | | | General | Math-1 | Math-2 | Math-3 | Code-1 | Code-2 | Code-3 |
|---|---|---|---|---|---|---|---|---|---|---|---|
| Uniform | - | - | - | - | 0.500 | 0.111 | 0.027 | 0.039 | 0.020 | 0.055 | 0.248 |
| Heuristic-2 | - | - | - | - | 0.400 | 0.194 | 0.024 | 0.017 | 0.052 | 0.098 | 0.216 |
| Heuristic-3 | - | - | - | - | 0.200 | 0.273 | 0.033 | 0.000 | 0.098 | 0.136 | 0.259 |
| RegMix | 224 | - | 112 | 2 | 0.426 | 0.336 | 0.020 | 0.012 | 0.030 | 0.074 | 0.092 |
| | 224 | - | 28 | 8 | 0.710 | 0.203 | 0.003 | 0.001 | 0.079 | 0.002 | 0.002 |
| | 448 | - | 56 | 8 | 0.503 | 0.330 | 0.020 | 0.001 | 0.077 | 0.009 | 0.059 |
| CLIMB | 224 | - | 112 | 2 | 0.481 | 0.366 | 0.018 | 0.008 | 0.033 | 0.034 | 0.060 |
| | 224 | - | 28 | 8 | 0.714 | 0.202 | 0.002 | 0.001 | 0.078 | 0.002 | 0.002 |
| | 448 | - | 56 | 8 | 0.417 | 0.422 | 0.027 | 0.011 | 0.079 | 0.008 | 0.035 |
| DeMix (Ours) | 211 | 30×7 | 56 | 0.01† | 0.017 | 0.187 | 0.406 | 0.017 | 0.135 | 0.217 | 0.022 |
| | 211 | 30×7 | 112 | 0.01 | 0.271 | 0.414 | 0.009 | 0.057 | 0.046 | 0.131 | 0.073 |
| | 212 | 30×7 | 224 | 0.01 | 0.218 | 0.403 | 0.002 | 0.063 | 0.044 | 0.176 | 0.094 |
| | 214 | 30×7 | 448 | 0.01 | 0.454 | 0.220 | 0.006 | 0.015 | 0.077 | 0.194 | 0.034 |
| | 219 | 30×7 | 896 | 0.01 | 0.181 | 0.430 | 0.004 | 0.135 | 0.029 | 0.149 | 0.071 |

† is the benchmarking cost derived from equal GPU-hour as in Table 1.

*Table 12.* Comparison between merged proxy models and models trained directly on mixed data. We use one math dataset $D_1$ and one code dataset $D_2$, and evaluate different $D_1/D_2$ mixture ratios. $\delta$ denotes the magnitude of parameter updates as defined in Eq. 4. Consistency is computed as the ratio between the merged-model score and the data-mix model score, averaged across the tested mixture ratios.

| Token Budget | $\delta$ | Type | $D_1/D_2$ Ratio | | | | | | | | Consistency | |
|---|---|---|---|---|---|---|---|---|---|---|---|---|
| | | | 20%/80% | | 40%/60% | | 60%/40% | | 80%/20% | | | |
| | | | Math | Code | Math | Code | Math | Code | Math | Code | Math | Code |
| 2B | 3.10% | Model merge | 5.79 | 21.39 | 6.26 | 20.09 | 7.17 | 17.76 | 8.38 | 15.94 | 1.04 | 0.97 |
| | | Data mix | 6.03 | 21.56 | 6.25 | 20.59 | 6.64 | 18.16 | 7.39 | 16.76 | | |
| 10B | 6.90% | Model merge | 6.49 | 25.12 | 9.48 | 22.50 | 11.55 | 18.05 | 14.57 | 14.52 | 0.96 | 0.81 |
| | | Data mix | 8.01 | 27.85 | 9.28 | 23.31 | 12.32 | 26.04 | 13.66 | 20.99 | | |
| 30B | 10.10% | Model merge | 8.09 | 30.06 | 11.17 | 23.09 | 15.34 | 19.56 | 19.95 | 15.42 | 0.82 | 0.75 |
| | | Data mix | 11.79 | 30.87 | 15.11 | 31.48 | 18.26 | 29.45 | 20.01 | 25.12 | | |
| 50B | 10.50% | Model merge | 8.48 | 33.19 | 11.88 | 25.50 | 15.74 | 20.97 | 20.87 | 14.31 | 0.79 | 0.75 |
| | | Data mix | 11.84 | 32.98 | 17.36 | 33.10 | 19.71 | 30.66 | 22.11 | 27.44 | | |

We conduct this comparison under different token budgets. The parameter-update magnitude $\delta$ is computed as defined in Eq. 4. We define the consistency score as the ratio between the merged-model score and the data-mix model score on the same benchmark, averaged across all tested mixture proportions. A consistency score closer to 1 indicates that the merged proxy model better matches the model trained directly on the corresponding mixed data.

The results are shown in Table 12. When $\delta$ is small, the merged proxy model exhibits high consistency with the data-mix model, suggesting that Eq. 5 is well satisfied in the small-update regime. As the token budget increases, $\delta$ naturally becomes larger, and the consistency moderately decreases. Nevertheless, even under relatively larger updates, the merged proxy model still preserves a reasonably high level of consistency, especially considering that it avoids training a separate model for every candidate mixture. These results provide additional empirical support for using merged proxy models to approximate mixed-data training during mixture search.

