# OpenReview forum: "Decouple Searching from Training: Scaling Data Mixing via Model Merging for Large Language Model Pre-training"
_ICML.cc/2026/Conference — ICML 2026 regular_

### Official Review · Reviewer_NUa3 · 2026-03-04

**Soundness:** 3
**Presentation:** 3
**Significance:** 3
**Originality:** 3
**Overall Recommendation:** 5
**Confidence:** 4

**Summary:**

This paper studies how to determine optimal dataset mixture ratios for LLM pre-training. Existing approaches either rely on unreliable tiny-scale proxies or require expensive large-scale exploration. The authors propose DeMix, which trains a separate component model on each dataset and then merges them with weights. These weights are interpreted as mixture ratio proxies. Experiments show that DeMix approximates real mixture rankings at lower total cost. The 22T-token DeMix Corpora is also released.

**Compliance With Llm Reviewing Policy:**

Affirmed.

**Final Justification:**

This paper ingeniously transforms model merging coefficients into task-specific ratios corresponding to the pre-training phase. The proposed approach is novel, and the authors have also open-sourced the mixed dataset constructed using this method. Therefore, I believe this paper deserves acceptance.

**Key Questions For Authors:**

As listed in Weakness 4, the conversion from merging weights to dataset mixture ratios depends fundamentally on the validity of Equation (5). However, this assumption may be influenced by dataset properties and training configurations. Could the authors provide further discussion on how the following factors affect the approximation: the size of candidate datasets, distributional differences between datasets, imbalance in each calibration dataset scale, the number of training iterations for component models, optimization strategies (e.g., learning rate schedule, optimizer type), the architecture of the component models.
Although the existing experiments are already well-developed, further discussion will provide more in-depth observations.

**Limitations:**

The authors discuss the impact statement but do not address its limitations, which are outlined in the Question and Weakness sections.

**Strengths And Weaknesses:**

**Strengths**

1. The idea of estimating optimal data mixture ratios through weighted model merging is novel and conceptually elegant. It reframes mixture search as a model-space optimization problem rather than a data-space exploration problem.

2. The release of a 22T-token pre-training corpus is a valuable contribution to the community and supports reproducibility and open research.

3. Extensive experiments demonstrate that DeMix can approximate real mixture rankings with reasonable fidelity, suggesting that model merging can serve as an effective and computationally efficient proxy for mixture search.

4. The paper is clearly written, well-structured, and easy to follow.

**Weaknesses**

1. The difference from MergeMix [1] is not clearly articulated and should be clarified. Since both approaches rely on model merging to optimize data mixtures, the conceptual and methodological distinctions should be more explicitly articulated.

    [1] MergeMix: Optimizing Mid-Training Data Mixtures via Learnable Model Merging.

2. In Table 2, the authors use seven component models to estimate mixture ratios. It remains unclear how sensitive the method is to the number of components. Would using more components improve mixture estimation? Would fewer components degrade performance?

3. The paper adopts 1.7B-parameter models as components, likely to control training cost. However, it is unclear how component scale affects the validity of the merging approximation. Is the effectiveness of DeMix invariant to component scale, or does scaling influence the fidelity of the proxy?

4. The mapping from merging weights to data mixture ratios relies critically on the assumption in Equation (5). However, the validity of this approximation likely depends on dataset scale and training dynamics. As shown in the last row of Figure 2, increasing the number of training tokens leads to a decrease in Spearman correlation but an increase in Capability Recovery. This phenomenon appears counterintuitive and requires further explanation.

---

> ### Author Rebuttal · Authors · 2026-03-30
>
> We sincerely thank Reviewer NUa3 for the highly constructive feedback. We address the main questions below and will incorporate the minor suggestions in the revised manuscript.
>
> **Q1: the difference from MergeMix**
> - MergeMix (available on arXiv after our submission) targets mid-training, whereas DeMix targets pre-training. This changes the optimization regime and the stability. Pre-training has larger parameter updates, making accurate proxy models harder to obtain. To solve this, DeMix introduces key optimizations like mixing 50% general-domain data for robustness.
> - Pretraining require more budget to achieve sufficient accuracy. Our component models were generally trained on 30B tokens, whereas MergeMix used only 5B. Therefore, we focus more on whether efficiency is improved and provide a detailed comparison with RegMix and Climb under varying budget settings.
> - The two works are complementary and both confirm that model merging reliably estimates data mixtures, implying it can be extended to other training stages.
> - We will cite and discuss this concurrent work in the revised version.
>
> **Q2: number of components**
>
> We refined data partitions from 7 to 15 (Table A). Conclusions remain consistent: DeMix's Spearman correlation with 30B/50B budgets matches the 12B-trained proxy, showing partition granularity minimally impacts performance.
>
> **Table A: Proxy accuracy with finer datasets partition**
>
> | Method | Budget | Avg. Spearman | Avg. Top 25% Spearman | Avg. Capability Recovery |
> | :--- | :--- | ---: | ---: | ---: |
> | Trained Proxy | 2*no. proxy | 0.66 | 0.41 | 0.77 |
> |  | 4*no. proxy | 0.81 | 0.41 | 0.81 |
> |  | 8*no. proxy | 0.81 | 0.48 | 0.84 |
> |  | 12*no. proxy | 0.84 | 0.58 | 0.87 |
> | DeMix | 2*15 | 0.44 | 0.26 | 0.78 |
> |  | 10*15 | 0.80 | 0.45 | 0.83 |
> |  | 30*15 | 0.83 | 0.59 | 0.86 |
> |  | 50*15 | 0.84 | 0.56 | 0.86 |
>
> **Q3: effect of model scaling**
> - We agree that component scale is an important question. Our goal is precisely to use efficient component models to guide larger-scale pre-training decisions; therefore, the most relevant test is whether mixtures selected on 1.7B transfer to larger models.
> - Table B shows that with the same mixture ratios, results of 8B models are consistent with that of 1.7B models: the ratios selected by DeMix achieve the best average rank among compared methods. This suggests that DeMix transfers well across scales.
>
> **Table B: Mixture performance with Qwen3-8B**
>
> | Method | no. proxy | general rank | code rank | math rank | avg rank |
> |---|---|---|---|---|---|
> | Uniform | | 7 | 54 | 64 | 41.67 |
> | Heuristic | | 64 | 18 | 37 | 39.67 |
> | RegMix | 112*2 | 30 | 56 | 31 | 39.00 |
> | | 28*8 | 15 | 53 | 56 | 41.33 |
> | | 56*8 | 10 | 41 | 51 | 34.00 |
> | CLIMB | 112*2 | 51 | 57 | 17 | 41.67 |
> | | 28*8 | 7 | 49 | 46 | 34.00 |
> | | 56*8 | 19 | 25 | 46 | 30.00 |
> | DeMix | 56 | 63 | 21 | 10 | 31.33 |
> | | 112 | 51 | 18 | 20 | 29.67 |
> | | 224 | 49 | 21 | 17 | 29.00 |
> | | 448 | 36 | 4 | 43 | **27.67** |
>
> **Q4: more training tokens reduce Spearman but improve Capability Recovery**
>
> More training improves model capability, so Capability Recovery increases. But it also enlarges parameter differences, which weakens the satisfaction of Eq. 5. Since Spearman measures ranking consistency, it is more sensitive to this proxy mismatch and can therefore decrease.
>
> **Key Q1: how other factors affect the effectiveness of Eq. (5)**
>
> Our existing and additional experiments during the rebuttal are able to demonstrate the following points. We will also discuss these in the revised version.
> - **Size of candidate datasets.** Increasing training tokens generally improves both the component models and the merged proxy model, making the proxy closer to the reference model in absolute capability. At the same time, larger datasets can increase parameter-space divergence, which may weaken the linear approximation behind Eq. (5). Empirically, this creates a trade-off.
> - **Distributional differences between datasets.** Our method remains effective on datasets with substantially different distributions, including both math and code data. (See Table A in our response to Reviewer xYcp for details.) This suggests that the method does not require all candidate datasets to be highly homogeneous.
> - **Imbalance in calibration dataset scale.** We use domain relative rankings across a wide range of sampled data mixtures to measure quality, minimizing evaluation bias as much as possible.
> - **Number of training iterations for component models.** We expect this factor to behave similarly to candidate dataset size: more training can improve absolute capability while also increasing divergence between component models.
> - **Optimization strategies and component model architecture.** We expect Eq. (5) to be more reliable when all component models share the same architecture and optimization pipeline. Our experiments follow this controlled setting using a standard training recipe and the widely used Qwen3 architecture.

---

> > ### Author Rebuttal · Reviewer_NUa3 · 2026-04-02
> >
> > Thank you for the response. I have no further questions.

---

> > > ### Author Response · Authors · 2026-04-07
> > >
> > > Thank you very much for your positive feedback and for taking the time to read our response carefully. We appreciate your acknowledgment that our concerns have been fully addressed, and we are grateful for your support of our paper.

---

### Official Review · Reviewer_MBtn · 2026-03-10

**Soundness:** 3
**Presentation:** 3
**Significance:** 3
**Originality:** 2
**Overall Recommendation:** 4
**Confidence:** 2

**Summary:**

DeMix constructs a training-free proxy through model merging, reducing the computational cost of LLM data mixture search and enabling larger-scale mixture exploration.

**Compliance With Llm Reviewing Policy:**

Affirmed.

**Final Justification:**

Since my concerns have been fully addressed, I maintain my positive stance on this paper.

**Key Questions For Authors:**

1. How the granularity and number of candidate datasets affect the search space and merging quality. Whether a coarser or finer partitioning of training data leads to better proxies?

**Limitations:**

yes

**Strengths And Weaknesses:**

### Strengths
1. The proxy paradigm shifts from "train a small model per mixture" to "merge pre-trained components," converting the per-proxy training cost into a one-time fixed cost.
2. Since proxies are constructed via weighted merging. This enables a much larger and more thorough exploration of the mixture space compared to training-based methods.
3. Dataset contribution: the release of DeMix Corpora with validated mixture ratios provides a directly usable resource for pre-training, filling a gap in the availability of high-quality public corpora with empirical data recipes.

### Weaknesses
1. The effectiveness of model merging as a proxy relies on shared base initialization and a small parameter update regime. It remains unclear whether these assumptions hold under different training conditions, such as longer training schedules, larger learning rates, or candidate datasets with highly divergent distributions.
2. Insufficient analysis of dataset granularity. The paper does not  explore how the granularity and number of candidate datasets affect the search space and merging quality. Whether a coarser or finer partitioning of training data leads to better proxies is an important practical question left unaddressed.
3. All experiments are conducted on the Qwen3-1.7B architecture. Whether the merging-based approximation holds at larger scales is an open question.

---

> ### Author Rebuttal · Authors · 2026-03-30
>
> We sincerely thank Reviewer MBtn for their meticulous review. Below, we directly address the three main concerns: robustness under different training conditions, the effect of dataset granularity, and transfer to larger model scales. We will incorporate the minor suggestions in the revised manuscript.
>
> **Q1: unclear whether assumptions hold under different training conditions**
>
> We agree that proxy fidelity depends on the magnitude of parameter updates. In practice, as training becomes longer or the learning rate increases, the approximation gradually weakens rather than failing abruptly. In our setting, the merged proxy is nearly identical to mixed-data training at 2B tokens; even at 50B tokens, the consistency remains around 80%, indicating that the proxy is still useful for ranking mixtures within the training budgets considered in this paper. Please see Table A in our response to Reviewer xYcp for details.
>
> **Q2 / Key Q1: Effect of finer partitioning of training data**
>
> - We performed an additional experiment by refining the data partition from 7 to 15 categories, and the main conclusion remains unchanged. Similar to Table 2 in the original paper, Table A provides a detailed results of DeMix’s proxy accuracy results. The conclusions are basically consistent across finer and coarser partitions. The Spearman correlation between the DeMix training budget for the 30B/50B component model and the 12B-trained proxy is approximately comparable, indicating that different partition granularities have little impact on the previous conclusions.
>
> **Table A: Proxy accuracy with finer datasets partition**
>
> | Method | Budget | Avg. Spearman | Avg. Top 25% Spearman | Avg. Capability Recovery |
> | :--- | :--- | ---: | ---: | ---: |
> | Trained Proxy | 2*no. proxy | 0.66 | 0.41 | 0.77 |
> |  | 4*no. proxy | 0.81 | 0.41 | 0.81 |
> |  | 8*no. proxy | 0.81 | 0.48 | 0.84 |
> |  | 12*no. proxy | 0.84 | 0.58 | 0.87 |
> | DeMix | 2*15 | 0.44 | 0.26 | 0.78 |
> |  | 10*15 | 0.80 | 0.45 | 0.83 |
> |  | 30*15 | 0.83 | 0.59 | 0.86 |
> |  | 50*15 | 0.84 | 0.56 | 0.86 |
>
> Regarding dataset granularity, finer partitioning has both benefits and costs:
>
> - Advantages of finer partitioning:
> 	- With a larger exploration space, the potential to yield better mixing ratios increases.
> - Disadvantages of finer partitioning:
> 	- (1) The search space grows exponentially. Compared with coarser partitioning, it lacks the prior information provided by merging similar datasets, so under the same search budget, it often performs worse than coarser partitioning;
> 	- (2) It requires more component models (DeMix) or more proxy training (RegMix/CLIMB);
> 	- (3) It can lead to many datasets with very small proportions. In data-mixing experiments, it becomes difficult to measure how fluctuations in their proportions affect final performance, making the mixing ratios of these small datasets less meaningful.
> - Therefore, in practical pretraining recipe design, overly fine partitioning is often avoided because it substantially enlarges the search space while making the effect of small-ratio datasets difficult to estimate [1,2].
>
> [1] Maximize your data’s potential: Enhancing llm accuracy with two-phase pretraining.
>
> [2]  Nemotron 3 nano: Open, efficient mixture-of-experts hybrid mambatransformer model for agentic reasoning.
>
> **Q3: Whether the merging-based approximation holds at larger scales**
>
> We supplements Table B with experiments on Qwen3-8B using the same mixture ratios as those in Table 3 of the original paper. The conclusions are similar to those for Qwen3-1.7B: the ratios produced by DeMix achieve better average rank, which provides supporting evidence that the method transfers beyond 1.7B and remains effective at 8B.
>
> **Table B: Mixture performance with Qwen3-8B**
>
> | Method | no. proxy | general avg | code avg | math avg | general rank | code rank | math rank | avg rank |
> |---|---|---|---|---|---|---|---|---|
> | Uniform | | 62.07 | 24.60 | 18.52 | 7 | 54 | 64 | 41.67 |
> | Heuristic | | 59.86 | 27.87 | 25.58 | 64 | 18 | 37 | 39.67 |
> | RegMix | 112*2 | 61.49 | 24.44 | 27.16 | 30 | 56 | 31 | 39.00 |
> | | 28*8 | 61.85 | 24.62 | 22.30 | 15 | 53 | 56 | 41.33 |
> | | 56*8 | 62.02 | 25.77 | 23.52 | 10 | 41 | 51 | 34.00 |
> | CLIMB | 112*2 | 60.82 | 23.98 | 29.80 | 51 | 57 | 17 | 41.67 |
> | | 28*8 | 62.10 | 25.10 | 24.50 | 7 | 49 | 46 | 34.00 |
> | | 56*8 | 61.78 | 26.96 | 24.25 | 19 | 25 | 46 | 30.00 |
> | DeMix | 56 | 60.32 | 27.61 | 34.14 | 63 | 21 | 10 | 31.33 |
> | | 112 | 60.81 | 27.80 | 28.94 | 51 | 18 | 20 | 29.67 |
> | | 224 | 60.83 | 27.36 | 29.68 | 49 | 21 | 17 | 29.00 |
> | | 448 | 61.19 | 30.60 | 25.02 | 36 | 4 | 43 | **27.67** |

---

> > ### Author Rebuttal · Reviewer_MBtn · 2026-04-02
> >
> > Thank you for the response. I maintain my positive stance on this paper.

---

> > > ### Author Response · Authors · 2026-04-07
> > >
> > > Thank you very much for your positive feedback and for taking the time to read our response carefully. We appreciate your acknowledgment that our concerns have been fully addressed, and we are grateful for your support of our paper.

---

### Official Review · Reviewer_xYcp · 2026-03-14

**Soundness:** 3
**Presentation:** 4
**Significance:** 3
**Originality:** 4
**Overall Recommendation:** 4
**Confidence:** 5

**Summary:**

The paper proposed an efficient way for proxy model estimation by model merging. In this way the proxy model doesn't require training but only testing to obtain performance score, which is further used to train a regression model to predict model performance. Compared with other methods for proxy model, it can achieve better results with lower budget.

**Compliance With Llm Reviewing Policy:**

Affirmed.

**Final Justification:**

My concerns have been adequately addressed through the rebuttal. The idea is sound with solid experiments to support. I will maintain my positive score

**Key Questions For Authors:**

See Weaknesses above

**Limitations:**

See Weaknesses above

**Strengths And Weaknesses:**

Strengths
- Use model merging to estimate the proxy model parameter and achieve good results on predicting model score
- The avg performance can beat other baseline methods

Weaknesses
- Is there any experiments proving equation 5 on proxy model estimation? Like what is the difference between the score of estimated proxy model and real proxy model score (actually use D1+D2 to train). And how the difference changes as proxy model size going up? From Figure 1, it shows that at 200B token budget, the Spearman stops going up, which may limits the proxy model when there are more budget.
- Does equation 5 requires independent between D1 and D2?
- From Table3, the avg rank improve is majorly due to Code and Math. The proxy model target is to minimize the average ranking score, why it shows bias towards Code and Math benchmarks
- In section 2.4, in each iteration, will the regressor trained from scratch or finetuned from  last iteration. If from scratch, only using scores from top ratios to train the regressor ( iteratively execute Step 2-4 ) may introduce bias into the regressor. For example it will not predict well on those low ratios.
- as the number of proxy model goes up, will the Spearman also become better?

---

> ### Author Rebuttal · Authors · 2026-03-30
>
> We sincerely thank Reviewer xYcp for the highly constructive feedback. We have addressed the main questions below and will incorporate the minor suggestions in the revised manuscript.
>
> **Q1.1: Experimental evidence supporting Eq. 5**
> - The validity of Eq. 5 is primarily grounded in prior works [1, 2]. In addition, we conduct experiments to further verify it: merged proxy model is consistent with mixed-data trained model especially under small parameter updates.
> - In detail, we select a math dataset as D1 and a code dataset as D2. We compare the performance of:
>     - A merged proxy model, obtained by merging two models trained separately on D1 and D2.
>     - A mixed-data trained model, trained directly on the combined dataset D1+D2.
> - Table A shows results across various token budgets and D1/D2 proportions. $\delta$ measures the magnitude of parameter updates as defined in Eq. 4. We define consistency as the ratio between the merged-model score and the mixed-data-trained model score on the same benchmark. With small $\delta$, the consistency is high, indicating that Eq. 5 is well satisfied. As the token budget increases, $\delta$ naturally increases, leading to a slight decrease in consistency. Notably, even when $\delta$ becomes relatively large, the consistency remains sufficiently high (~80%).
> - While we have only verified this trend at the 1.7B scale, the results provide supporting evidence in other settings.
>
> **Table A: comparison of merged proxy model and mixed-data model**
> | Token budget | $\delta$ | Type | 20/80 Math Avg. | 20/80 Code Avg. | 40/60 Math Avg. | 40/60 Code Avg. | 60/40 Math Avg. | 60/40 Code Avg. | 80/20 Math Avg. | 80/20 Code Avg. | Math consistency | Code consistency |
> | :---: | :---: | :---: | :---: | :---: | :---: | :---: | :---: | :---: | :---: | :---: | :---: | :---: |
> | 2B | 3.10% | model merge | 5.79 | 21.39 | 6.26 | 20.09 | 7.17 | 17.76 | 8.38 | 15.94 | 1.04 | 0.97 |
> | | | data mix | 6.03 | 21.56 | 6.25 | 20.59 | 6.64 | 18.16 | 7.39 | 16.76 | | |
> | 10B | 6.90% | model merge | 6.49 | 25.12 | 9.48 | 22.5 | 11.55 | 18.05 | 14.57 | 14.52 | 0.96 | 0.81 |
> | | | data mix | 8.01 | 27.85 | 9.28 | 23.31 | 12.32 | 26.04 | 13.66 | 20.99 | | |
> | 30B | 10.10% | model merge | 8.09 | 30.06 | 11.17 | 23.09 | 15.34 | 19.56 | 19.95 | 15.42 | 0.82 | 0.75 |
> | | | data mix | 11.79 | 30.87 | 15.11 | 31.48 | 18.26 | 29.45 | 20.01 | 25.12 | | |
> | 50B | 10.50% | model merge | 8.48 | 33.19 | 11.88 | 25.5 | 15.74 | 20.97 | 20.87 | 14.31 | 0.79 | 0.75 |
> | | | data mix | 11.84 | 32.98 | 17.36 | 33.1 | 19.71 | 30.66 | 22.11 | 27.44 | | |
>
> **Q1.2: Spearman saturation at 200B tokens constrains proxies**
>
> We agree that DeMix exhibits saturation beyond a certain budget. Theoretically, baseline methods like RegMix and CLIMB can infinitely approach the ground truth by increasing the training budget (e.g., 40B real-mixture proxy experiment as a surrogate for a 50B target pretraining). However, such scaling reduces the practical value of the proxy, whose purpose is to enable efficient mixture search. DeMix’s key advantage is therefore its effectiveness under realistic, limited budgets, where it consistently outperforms RegMix and CLIMB.
>
> **Q2: Does Eq. 5 requires independent between D1 and D2?**
>
> Strictly speaking, yes—the derivation of Eq. 5 in prior work [1, 2] assumes independence between D​1 and D2. Table A shows this holds well for low-correlation tasks. We also observe that the assumption is weaker when the datasets shares similar distribution. In Demix, the assumption is not exact, but it is a reasonable approximation when the component datasets are weakly correlated. A deeper theoretical exploration will be reserved for future work.
>
> **Q3: bias towards code and math benchmarks**
>
> Our setup does not intentionally favor code or math. Both DeMix and the baselines optimize the average ranking across general, code, and math tasks. The larger gains on code/math mainly stem from the pretrained model already being strong on general tasks, leaving more room for improvement on code and math.
>
> **Q4: regressor trained from scratch or finetuned from last iteration**
>
> In each iteration, the data collected in previous rounds are combined with the new data, and the regressor is retrained from scratch using the accumulated dataset. This approach draws on CLIMB. As we aim to find the optimal mixture, low ratios require no special attention. We apologize for any ambiguity and will revise the description.
>
> **Q5: Will the Spearman also become better as the number of proxy model goes up?**
>
> Not necessarily. Spearman measures the intrinsic quality of the proxy model. Increasing the number of proxy models does not improve the Spearman itself, but provides a more precise estimate of it. We believe 96 proxy models in our experiments are sufficient.
>
> [1] Shadow-ft: Tuning instruct via base.
>
> [2] Knowledge is a region in weight space for fine-tuned language models.

---

> > ### Author Rebuttal · Reviewer_xYcp · 2026-04-03
> >
> > Thank authors for the response. I will maintain my positive stance on this paper.

---

> > > ### Author Response · Authors · 2026-04-07
> > >
> > > Thank you very much for your positive feedback and for taking the time to read our response carefully. We appreciate your acknowledgment that our concerns have been fully addressed, and we are grateful for your support of our paper.

---

### Decision · Program_Chairs · 2026-04-30

**Decision:**

Accept (regular)

**Comment:**

This is a paper for data mixture selection. As usual with such techniques, the goal is to optimize the data mixture in such a way that models are better trained while avoiding paying a large computational overhead for performing the optimization. The authors take the proxy model approach and reduce cost by using a type of model merging. They show that for certain budgets, this produces superior performance to certain baselines.

The reviewers were broadly positive; they had questions on the validity of the approach. The authors did a solid job providing further evidence for these aspects on their technique.

The main outstanding issues here (and also spotted by some of the  reviewers) are that the authors do not provide a huge amount of evidence for their technique, nor compare it to potential other competitors. There are basically two baselines in the paper, even though this is a huge space that has lots of comparable approaches. For example, DOGE (Fan et al ’24) is a really popular approach that should fall within the authors’ scope. The benchmarks and datasets used are fairly limited as well.

More generally on baselines, the authors directly remove a whole bunch of baselines by simply claiming they are “inferior”: “We exclude earlier inferior methods, including DoReMi (Xie et al., 2023) and Rho Loss (Mindermann et al., 2022), as they depend on evaluation loss instead of proxies.” This is a big claim that requires justification. The authors try to do some of this work by listing some potential limitations of DoReMi in the introduction, but this is pretty unconvincing. There are certainly versions that can address target domains that differ from training domains. As one example, Skill-it! (Chen et al ’23), which the authors cite but do not discuss or compare against, can handle this scenario.

I think this is a nice work—with a solid underlying idea, and solid results—but that could potentially use more evidence in support of its claims.